# MicroRNA-766-3p Contributes to Anti-Inflammatory Responses through the Indirect Inhibition of NF-κB Signaling

**DOI:** 10.3390/ijms20040809

**Published:** 2019-02-14

**Authors:** Kunihiro Hayakawa, Mikiko Kawasaki, Takuya Hirai, Yuko Yoshida, Hiroshi Tsushima, Maki Fujishiro, Keigo Ikeda, Shinji Morimoto, Kenji Takamori, Iwao Sekigawa

**Affiliations:** 1Institute for Environment and Gender-Specific Medicine, Juntendo University Graduate School of Medicine, Chiba 279-0021, Japan; mikidx@nifty.com (M.K.); thirai@juntendo.ac.jp (T.H.); yyoshida@musashino-u.ac.jp (Y.Y.); htsushi@juntendo.ac.jp (H.T.); mfujishi@juntendo.ac.jp (M.F.); ktakamor@juntendo.ac.jp (K.T.); isekigawa@mva.biglobe.ne.jp (I.S.); 2Department of Internal Medicine and Rheumatology, School of Medicine, Juntendo University, Tokyo 113-8421, Japan; 3Department of Internal Medicine and Rheumatology, Juntendo University Urayasu Hospital, Chiba 279-0021, Japan; keigo@juntendo.ac.jp (K.I.); morimoto@juntendo.ac.jp (S.M.)

**Keywords:** microRNA (miRNA), tumor necrosis factor-α (TNF-α), interleukin(IL)-1β, inflammation, nuclear factor-κB (NF-κB), rheumatoid arthritis (RA), abatacept, mineralocorticoid receptor

## Abstract

MicroRNA (miRNA) is small RNA of 20 to 22 nucleotides in length and is stably present in plasma. Regulating the expression of miRNA taken into cells has been suggested as a general therapeutic approach. We identified the novel anti-inflammatory miRNA hsa-miR-766-3p and investigated its biological function in human rheumatoid arthritis (RA) fibroblast-like synoviocyte MH7A cells. To verify the function of the miRNA present in the plasma of RA patients, we performed a comprehensive analysis of the miRNA expression during abatacept treatment and identified eight miRNAs with significantly altered expression levels. Among these eight miRNAs, miR-766-3p was found to have a clear function. The expression of inflammatory genes in response to inflammatory stimuli was suppressed in MH7A transduced with miR-766-3p. We showed that miR-766-3p indirectly reduced the activation of NF-κB and clarified that this mechanism was partially involved in the reduction of the mineralocorticoid receptor expression. In addition, the inflammatory responses were suppressed in other types of cells. These results indicate the novel function of miR-766-3p, findings that may aid in the development of therapies to suppress inflammation, not only in RA but also in other diseases.

## 1. Introduction

MicroRNA (miRNA) is a non-coding RNA with a chain length of approximately 22 nucleotides and is known to control gene expression [1]. In recent years, it has been reported that miRNA circulates in the blood [2,3,4]; thus, circulating miRNA has received a great deal of attention, as these miRNAs are useful as biomarkers of disease [1,3]. However, the function of such blood-circulating miRNAs is unclear.

Rheumatoid arthritis (RA) is a systemic autoimmune inflammatory disease that is associated with synovial inflammation and bone and cartilage destruction [5]. Synoviocytes have been shown to act as the primary pro-inflammatory effector cells in RA. In response to pro-inflammatory mediators, most notably tumor necrosis factor-α (TNF-α), synoviocytes express tissue-degrading enzymes, such as matrix metalloproteinases (MMPs), and inflammatory cytokines and chemokines [6]. These inflammatory mediators from synoviocytes support the activation and differentiation of infiltrating immune cells [7].

In this study, we investigated the changes in blood-circulating miRNAs in RA patients before and after treatment with abatacept, a biologic agent used for RA that consists of an extracellular domain of human cytotoxic T-lymphocyte-associated antigen 4 (CTLA-4) and the Fc portion of human immunoglobulin G1. Using the synovial cell line, we investigated the function of blood-circulating miRNAs, a type of miRNA that demonstrates significant fluctuations. In addition, we tried to determine whether this effect was observed not only in synovial cells, but also in other cells. Finally, we attempted to elucidate whether this miRNA acts as an anti-inflammatory miRNA in various diseases.

## 2. Results

### 2.1. Changes in Plasma miRNA Levels before and after Abatacept Treatment in Patients with RA

We analyzed the plasma miRNA levels before and after abatacept treatment in patients with RA using a miRNA array (Table 1). We chose miRNAs with significantly altered expression levels (increased or decreased) based on their mean fluorescence intensity because they met the following criteria: 1) large variation in expression, 2) stable array results (stable number of detected specimens), and 3) measurement possible. We therefore focused on eight miRNAs, hsa-miR-625* (625-3p), 766 (766-3p), 1203, 1225-5p, 4259, 4299, 4505, and 4739.

### 2.2. Suppression of TNF-Induced Inflammatory Responses by hsa-miR-766-3p

In a previous study, blood-circulating miRNA was shown to have a distinct function while also playing a role in various pathogenies and to regulate the gene expression in other tissues [8,9]. We therefore hypothesized that these blood circulating miRNAs might be involved in RA pathology. Using an MH7A human synovial fibroblast cell line, eight miRNA mimics were screened for changes in their TNF-α-induced inflammatory gene expression. miRNA-transfected MH7A cells were treated with TNF-α, and the expression of interleukin (IL)-1β, IL-6, IL-8, and MMP3 mRNAs was analyzed by quantitative polymerase chain reaction (qPCR). Through this screening, we selected hsa-miR-766-3p, as it suppressed all of the investigated TNF-α-induced inflammatory genes by more than 30% (Figure 1).

### 2.3. Blunted Induction of Inflammatory Responses in miR-766-3p-Treated MH7A Cells

Using miR-766-3p mimic, we further examined the inhibitory potential of this miRNA. The suppressive effect of miR-766-3p in cells stimulated by TNF-α was dose-dependent (Figure 2A). We then examined the effects of miR-766-3p on the induction of inflammatory genes by TNF-α and IL-1β. Transfection of the cells with miR-766-3p mimic significantly diminished the induction of these genes (Figure 2B,C). However, the IL-1β expression was not suppressed in miR-766-3p-transfected cells (Figure 2C). There is a difference in the signal transduction between IL-1β and TNF-α, so miR-766-3p may affect this difference in MH7A cells. A formazan assay showed that, in this experimental setting (exposure to cytokines 48 h after transfection), the number of viable cells was slightly decreased among miR-766-3p-treated cells (Figure 2D). To further confirm the effect on complex inflammatory stimuli, to which they would be exposed in vivo, miR-766-3p-transfected MH7A cells were co-cultured with peripheral blood mononuclear cells (PBMCs) from healthy volunteers with or without lipopolysaccharide (LPS). Activated PBMCs produce an array of molecules that affect the inflammatory response (e.g., TNF-α and IL-1β) [10]. A qPCR showed that the expression of IL-1β, IL-6, IL-8 and MMP3 was markedly induced in MH7A cells after exposure to PBMCs with LPS. Among these genes, the induction of IL-6 and MMP3 was significantly reduced in miR-766-3p-transfected MH7A cells (Figure 2E).

These results suggested that miR-766-3p blunts the responses to inflammatory stimuli, particularly those to TNF-α, in MH7A cells. In addition, miR-766-3p may regulate different molecules in TNF-α and IL-1β signaling.

Next, we examined how the expression of miR-766-3p was induced, based on the suspicion that the expression was increased by inflammatory stimulation. However, miR-766-3p was not detected in LPS-stimulated PBMCs (data not shown), and expression in MH7A cells was not increased by inflammatory stimuli (Figure 2F). In addition, we used S-TuD (a miRNA inhibitor) to investigate the involvement of endogenous miRNA. However, it did not promote inflammatory responses (Figure 2G), making it unlikely that intracellular miR-766-3p typically participates in anti-inflammatory mechanisms. Thus, for miR-766-3p to exhibit an anti-inflammatory effect in MH7A cells, miRNA needs to be taken up from extracellular sources.

### 2.4. Involvement of miR-766-3p in the Suppression of Cytokine-Induced NF-κB Activation

Previous reports showed that the cytokine-induced IL-6 or IL-8 expression was dependent on NF-κB in MH7A cells [11]. The suppression of cytokine-induced inflammatory genes by miR-766-3p may be caused by the inhibition of NF-κB activation. To examine our hypothesis, reporter assays were performed. MH7A cells were transiently co-transfected with pGL4.32 (pNF-κB-Luc2P) along with miRNA mimics. The cells were treated with TNF-α or IL-1β and subjected to a luciferase assay. As shown Figure 3A, treatment with TNF-α or IL-1β markedly induced the activation of NF-κB activity, and this activity was reduced in miR-766-3p-transfected MH7A cells by approximately 27% under TNF-α stimulation and approximately 16% under IL-1β stimulation at 6 h and by approximately 32% under TNF-α stimulation and approximately 20% under IL-1β stimulation at 24 h. These results indicated that miR-766-3p partially suppressed the cytokine-induced activation of NF-κB. On the other hand, in comparison to negative control (NC)-miRNA-induced MH7A cells, miR-766-3p-transfected MH7A cells showed no change in the translocation of NF-κB subunit p65 into the nucleus or its binding to the κB sites after inflammatory stimuli (Figure 3B–D). 

Besides NF-κB, AP-1—which regulates inflammatory processes—is another important transcription factor in inflammatory cytokine signaling [12]. We examined the activation of AP-1 in cytokine-treated MH7A cells by reporter assays. However, cytokine-induced AP-1 activation was not observed in MH7A cells (Figure 3 E).

### 2.5. Mineralocorticoid Receptor Contributes to the miR-766-3p-Induced Anti-Inflammatory Response

The mineralocorticoid receptor (MCR; NR3C2) is a direct target of hsa-miR-766-3p [13] (Figure 4A,B). In addition, several reports have shown an association between MCR and inflammatory responses [14,15]. Thus, we examined the association between MCR and NF-κB activation after TNF-α or IL-1β stimulation in MH7A cells. For this purpose, we performed loss-of-function studies of MCR using small interfering RNA (siRNA). siRNA-transfected MH7A cells showed the reduced expression of MCR mRNA and protein (Figure 4C,D). We transiently co-transfected MH7A cells with pNF-κB-Luc and siRNAs. After transfection, we treated the cells with TNF-α or IL-1β for 6 h and subjected them to a luciferase assay to evaluate the NF-κB activity. The knockdown of endogenous MCR significantly reduced the NF-κB activation by inflammatory stimuli in MH7A cells (Figure 4E). The translocation of p65 into the nucleus and its binding to the κB sites after inflammatory stimuli had no effect, similar to miR-766-3p transfection (Figure 4F,G). In addition, to investigate whether the canonical signal of MCR is involved in this suppression mechanism, pNF-κB-Luc transiently transfected MH7A cells were treated with inflammatory cytokines and aldosterone (ALD; MCR agonist) or eplerenone (EPL; MCR antagonist). The result showed that cytokine-induced NF-κB activation was not modified by ALD and EPL (Figure 4I). These data suggest that NF-κB activation is suppressed by the direct reduction of MCRs by miR-766-3p, and that this mechanism is not involved in the canonical MCR signaling pathway.

### 2.6. Generalizability of the Suppression of Inflammatory Cytokines in mR-766-3p-Transfected Cells

To examine whether or not the suppression of inflammatory responses by miR-766-3p was specific to MH7A cells, we tested other cell types, including the human chondrocyte cell line C28/I2 and primary normal human mesangial cells (NHMCs). C28/I2 cells and NHMCs were transfected with miR-766-3p mimics, and two days later, cells were exposed to TNF-α or IL-1β for 24 h. A qPCR revealed that, in C28/I2 cells, the expression of the inflammatory genes CCL2, COX2 and MMP13 was induced by TNF-α and IL-1β in NC-miR-transfected cells, while the induction was slightly reduced or not reduced at all in miR-766-3p-transfected cells (data not shown). A qPCR of NHMCs revealed that, similarly to MH7A cells, the inflammatory gene expression was reduced in miR-766-3p-transfected cells, except for the IL-1β-induced CCL2 expression (Figure 5A).

The cytokine-inducible CCL2 expression was dependent on NF-κB [16], whereas the constitutive expression of CCL2 was dependent on NF-κB and AP-1 [17]. Thus, we examined the NF-κB activity using a reporter assay. The data showed that the NF-κB activity was reduced in each miR-766-3p-transfected cell line, similarly to MH7A cells (Figure 5B,C). These results suggest that the suppression of NF-κB signaling by miR-766-3p is a general phenomenon, while the expression of inflammatory genes in C28/I2 cells and NHMCs may be controlled by other transcription factors or signaling, thus explaining the lack of any observed changes.

## 3. Discussion

The kinetics of blood circulating miRNAs have been reported to several diseases. Previously, we reported changes in plasma miRNAs in a patient with polymyositis and dermatomyositis [4]. On the other hand, these only showed the possibility of applying plasma miRNAs as biomarkers, and few reports have focused on their function. In this study, we hypothesized that, among the blood circulating miRNAs that fluctuate before and after abatacept treatment in RA patients, there are miRNAs that control inflammation. Using miRNA array analysis, we narrowed this down to eight possible miRNAs, and we revealed that hsa-miR-766-3p contributed to the regulation of the inflammatory response.

hsa-miR-766-3p has been reported to be a biomarker for cancer [18,19,20] and post-exercise change [21,22]. This miRNA also contributes to cancer cell proliferation [20,23] as well as the suppression of metastasis [24]. In acute promyelocytic leukemia cells, the expression of miR-766-3p was elevated, while that of BAX, a pro-apoptotic protein, was suppressed [25]. The expression of miR-766-3p has been reported to increase due to aging [26]. In addition, miR-766-3p is a miRNA that is highly expressed in inflamed pulp [27]. With regard to immune diseases, the miR-766-3p levels in the serum of systemic lupus erythematosus patients with renal disorder have been found to be decreased in comparison to patients without renal disorder, suggesting that this miRNA may play a pivotal role in the PI3K-AKT-mTOR pathway [28]. However, the suppression of inflammatory responses by miR-766-3p has not been reported.

In the present report, we described the novel anti-inflammatory effects of miR-766-3p in several cell lines. We found that miR-766-3p reduced the induction of inflammatory genes in MH7A cells by TNF-α, IL-1β and LPS-activated PBMCs (Figure 2). Furthermore, it was revealed that miR-766-3p reduced the NF-κB activity, suggesting that it is responsible for inhibiting the inflammatory gene expression (Figure 3). Furthermore, it was clarified that the downregulation of MCR by miR-766-3p is involved in this inhibition mechanism (Figure 4). However, in miR-766-3p-induced MH7A cells, there are no change in the translocation of the p65 subunit into the nucleus and its binding to the κB sites after inflammatory stimuli, the precise mechanism of repression by miR-766-3p has not been elucidated. After the NF-κB subunits are translocated into the nucleus, they are subjected to molecular modification, such as methylation, acetylation and phosphorylation. Subsequent histone modification and chromatin remodeling allows binding to the NF-κB subunit at the κB sites [29]. The expression of its target genes is enhanced according to these cascades. Thus, there is a possibility that miR-766-3p and the MCR are involved after such transcriptional regulation.

In this study, we investigated the miRNAs whose plasma expression changed after abatacept treatment in RA patients, and focused on miR-766-3p, as its expression was markedly higher after treatment. We investigated the tissues in which miR-766-3p was produced under inflammatory conditions and found that it seemed not to be produced in the PBMCs or synovium (Figure 2F). Furthermore, although the influence of endogenous miR-766-3p was examined, the inflammatory response was nearly unchanged by inhibition of intracellular miR-766-3p (Figure 2G). Thus, our data suggested that the suppression of inflammatory responses is exerted when miR-766-3p is extracellularly present in large amounts and is taken into the cell. In addition, exosome containing miRNAs secreted from mesenchymal stem cells reportedly suppress cell migration and angiogenesis in human cells and collagen-induced arthritis in mice [30]. Human mesenchymal stem cells may therefore be a potential source of miR-766-3p.

In the present investigation, we showed that MH7A cells, C28/I2 cells and NHMCs can acquire resistance to inflammation by miR-766-3p transfection, suggesting that if miRNA can be efficiently injected into target cells, it may be useful for inflammatory disease therapy.

## 4. Materials and Methods 

### 4.1. Clinical Specimens

Plasma samples were obtained from 10 patients with RA before and after abatacept treatment for use in a miRNA array (Table 2). These patients showed a good prognosis after abatacept treatment. Ethical approval for this study was granted by the institutional review board of Juntendo University Urayasu Hospital (IRB no. 2009-009). Written informed consent was obtained from all of the patients according to the Declaration of Helsinki.

### 4.2. miRNA Array Analyses

Total RNA was extracted from 300 μL of plasma obtained from RA patients before and 3 months after treatment with abatacept using a 3D-gene RNA extraction kit (TORAY, Tokyo, Japan) and miRNeasy mini kit (Qiagen, Hilden, Germany). The extracted RNA was labeled using a miRCURY LNA microRNA Hy5 Power labeling kit (Exiqon, Vedbaek, Denmark), and the labeled targets were then hybridized to a 3D-Gene Human miRNA 4-plex chip (V17_V1.0.0, TORAY). Hybridization images were scanned using a GenePix4400A device (Molecular Devices, Sunnyvale, CA, USA), and the miRNA expression was assessed based on the signal intensity calculated as the median of the foreground signal minus the mean of the negative control signals + 2 standard deviations. As there was no definite internal control for plasma miRNA, the median intensity levels were calculated using the per-chip 95th percentile method [4].

### 4.3. Reagents

miRNA mimics were purchased from GeneDesign (Osaka, Japan). The NC-miRNA mimic sequence was UACUGAGAGACAUAAGUUGGUC [31]. Human recombinant TNF-α and human recombinant IL-1β were purchased from R&D Systems (Minneapolis, MN, USA). LPS (*Escherichia coli* 0111; B4) and aldosterone were purchased from Sigma-Aldrich Japan (Tokyo, Japan). Eplerenone was purchased from Selleck Biotech (Osaka, Japan). siRNA targeting to NR3C2 (SMARTpool:ON-TARGET*plus* NR3C2 siRNA) and non-targeting control siRNA (ON-TARGET*plus* Non-targeting Pool) were purchased from Dharmacon (Horizon Discovery; Cambridge, UK).

### 4.4. Cells

The human synovial fibroblast cell line MH7A was obtained from Riken Cell Bank (Ibaraki, Japan) [32], the human chondrocyte cell line C28/I2 was obtained from Merck Millipore (Darmstadt, Germany), and normal human mesangial cells (NHMCs) were obtained from Lonza (Basel, Switzerland). MH7A cells were maintained in RPMI-1640 (Sigma-Aldrich), C28/I2 cells were maintained in Dulbecco’s Modified Eagle’s Medium (DMEM)-high glucose (Sigma-Aldrich), and NHMCs were cultured in DMEM/Ham’s F-12 (Wako, Osaka, Japan). Medium containing 1% fetal bovine serum was generally used for studies.

### 4.5. Transient Transfectants

Using Lipofectamine 3000 (Thermo Fisher Scientific, Waltham, MA, USA), MH7A, C28/I2, and NHMCs were transfected with miRNA mimics (0.1–30 nM) or siRNA (20 nM) according to the manufacturer’s instructions. After performing overnight incubation twice, the cells were stimulated with TNF-α (10 ng/mL) or IL-1β (10 ng/mL) for 0.5-24 h and were then subjected to a real-time qPCR, western blotting or a formazan assay.

In some experiments, cells were co-transfected with 1–10 μg of pGL4.32[luc2P/NF-κB-RE/Hygro] Vector (Promega, Madison, WI, USA) or pGL4.44[luc2P/AP1 RE/Hygro] Vector (Promega). After incubation for 24 h, the cells were seeded in 96-well plates. After incubation overnight, the cells were exposed to TNF-α or IL-1β for 6 or 24 h and then subjected to a luciferase assay. The luciferase assay was performed using the Bright-Glo Luciferase Assay System (Promega) according to the manufacturer’s instructions.

### 4.6. Co-Culture of MH7A with PBMCs

PBMCs obtained from healthy volunteers were isolated using Ficoll (GE Healthcare, Buckinghamshire, UK). miRNA-transfected MH7A cells were co-cultured with PBMCs using Transwell inserts (Corning, Corning, NY, USA) with a 0.4-μm porous membrane and then treated with LPS (1 μg/mL) for 24 h. After incubation, MH7A cells were subjected to qPCR.

### 4.7. RNA Extraction and qPCR

Total RNA was extracted using an miRNeasy Mini kit or RNeasy Mini kit (Qiagen) according to the manufacturer’s instructions and reverse transcribed with a PrimeScript RT reagent Kit (TaKaRa, Shiga, Japan). The qPCR was performed with a TB Green Premix Ex Taq II (TaKaRa) and the LightCycler 480 System II (Roche, Basel, Switzerland). The results were normalized to the β-actin expression. The primer sequences for *IL1B*, *IL6*, *CXCL8*, *MMP3*, *NR3C2*, *CCL2*, *COX2*, and *ACTB* are available on request.

Quantification of miRNA was performed using the TaqMan MicroRNA Reverse Transcription Kit (Thermo Fisher Scientific) and THUNDERBIRD Prove qPCR Mix (TOYOBO, Osaka, Japan) and the relative hsa-miR-766-3p expression levels after normalization to U6 small nuclear RNA.

### 4.8. Western Blotting

Cells were lysed with RIPA buffer (BioDynamics Laboratory, Tokyo, Japan) containing protease inhibitor cocktail (Roche) and phosphatase inhibitor cocktail (Thermo Fisher Scientific). Nucleic proteins were extracted from MH7A cells using a NE-PER™ Nuclear and Cytoplasmic Extraction Reagents (Thermo Fisher Scientific).

The following antibodies were used for western blotting: anti-p65 (D14E12; Cell signaling, Danvers, MA, USA), anti-histone H3 (ab1791; Abcam, Cambridge, MA), anti-NR3C2 (ab64457; Abcam), and anti-β-actin (AC-15; Sigma-Aldrich). Horseradish peroxidase-conjugated anti-IgG secondary antibodies against rabbit IgG (Dako, Glostrup, Denmark) or mouse IgG (Cell signaling) were used with Chemi-Lumi One substrate (Nacalai Tesque, Kyoto, Japan). A densitometric analysis was performed using the ImageJ software program (Rasband, W.S., Image J, U.S. National Institutes of Health, Bethesda, MD, USA; http://rsb.info.nih.gov/ij/).

### 4.9. Formazan Assays

The number of viable cells was assessed by a formazan assay using a Cell Counting Kit-8 (Dojindo Laboratory, Kumamoto, Japan) according to the manufacturer’s instructions.

### 4.10. Evaluation of DNA Binding Activity of NF-κB.

The DNA binding activity of NF-κB p65 subunit was evaluated using an ELISA-based TransAM NF-κB p65 Transcription Factor Assay Kit (Active Motif, Carlsbad, CA, USA), as described previously [33]. Five micrograms of nuclear extract protein were used for the assay.

### 4.11. Statistical Analyses

miRNA array data were analyzed using Microsoft Excel (Microsoft, Redmond, WA, USA). Paired *t* test was used to compare samples taken from patients before and after treatment.

Statistical analyses of in vitro studies were performed with the GraphPad Prism 6 software program (GraphPad Software, La Jolla, CA, USA). Data are expressed as the mean ± standard error of the mean. Statistical analyses were performed using the non-parametric Mann-Whitney *U* test to compare data in different groups. *P* values of < 0.05 were considered to indicate a statistically significant difference.

## Figures and Tables

**Figure 1 ijms-20-00809-f001:**
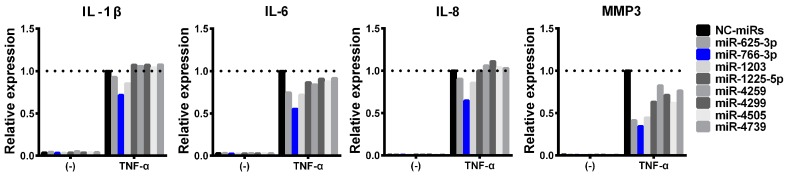
Effects of miRNA mimics on the induction of inflammatory responses by TNF-α. MH7A cells (a human synovial fibroblast cell line) were transfected with miRNA mimics (hsa-miR-625-3p, hsa-miR-766-3p, hsa-miR-1203, hsa-miR-1225-5p, hsa-miR-4259, hsa-miR-4299, hsa-miR-4505, hsa-miR-4739, or negative control [NC] miRNA) (5 nM). After performing overnight incubation twice, cells were treated with TNF-α (10 ng/mL) for 24 h, and the expression of the indicated genes was evaluated by a qPCR. Assays were performed in duplicate. Representative data are shown.

**Figure 2 ijms-20-00809-f002:**
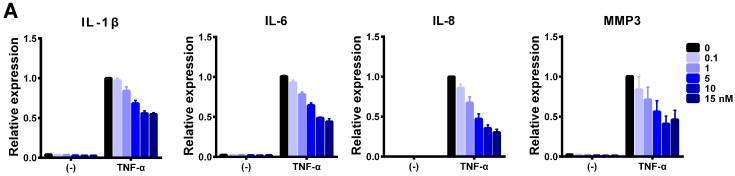
Suppression of cytokine-induced inflammatory genes by miR-766-3p. (**A**) MH7A cells were transfected with the indicated concentrations of miR-766-3p mimic and exposed to TNF-α (10 ng/mL) for 24 h. The expression of IL-1β, IL-6, IL-8, and MMP3 was determined by a qPCR. (**B**–**D**) Cells were transfected with negative control miRs (NC-miRs) or miR-766-3p (5 nM). After incubation, cells were stimulated by (**B**,**D**) TNF-α or (**C**,**D**) IL-1β (10 ng/mL) for 24 h and subjected to (**B**,**C**) a qPCR or (**D**) a formazan assay. (**E**) miRNA mimic-transfected MH7A cells were co-cultured with peripheral blood mononuclear cells (PBMCs) using a Transwell system. Cells were treated with lipopolysaccharide (LPS; 1 μg/mL) for 24 h, and MH7A cells were subjected to a qPCR. The expression of the indicated genes was normalized to that of *ACTB* and then normalized to the respective values in TNF-α-, IL-1β- or PBMC + LPS-stimulated NC-miR-transfected cells. (**F**) MH7A cells were treated with TNF-α for 24 h. The expression of hsa-miR-766-3p was determined by a qPCR, and normalized to that of U6 small nuclear RNA. (**G**) MH7A cells were transfected with miRNA mimics. After incubation for 24 h, additional transfection with NC-S-TuD or 766-S-TuD (inhibitor of hsa-miR-766-3p), and after incubation for 24 h and treatment with TNF-α for 24 h. The cells were then subjected to a qPCR. Assays were performed in quadruplicate (**A**–**E**), or duplicate (**F**,**G**). Data are expressed as the mean ± SEM. Asterisks indicate statistically significant differences (*p* < 0.05).

**Figure 3 ijms-20-00809-f003:**
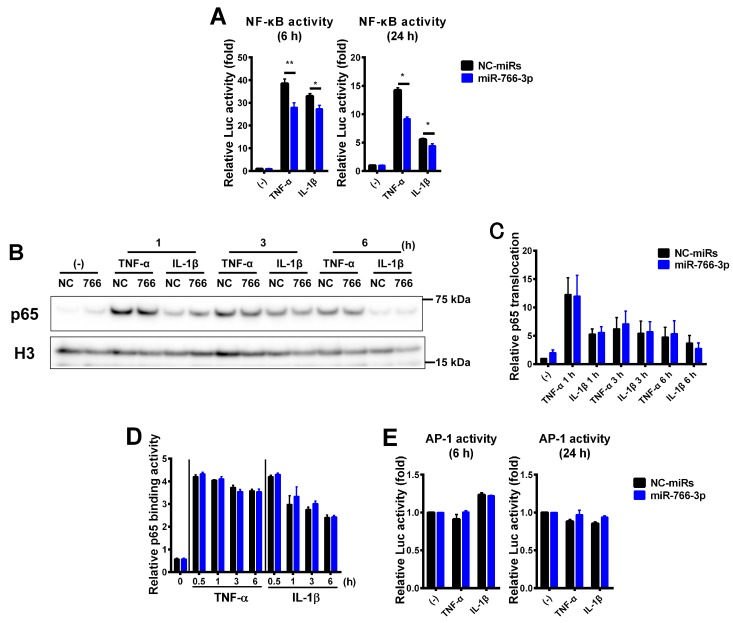
Suppression of cytokine-induced nuclear factor-κB (NF-κB) activation by miR-766-3p. (**A**) MH7A cells were co-transfected with pNF-κB-Luc along with miRNA mimics and were then exposed to TNF-α or IL-1β for 6 or 24 h. Cells were then subjected to a luciferase assay to evaluate the activity of NF-κB. The luciferase activity was normalized by the number of viable cells and then normalized to the respective values in the vehicle samples. Assays were performed in sextuplicate or quadruplicate, and data are expressed as the mean ± SEM. Asterisks indicate a statistically significant difference (*, *p* < 0.05; **, *p* < 0.01). (**B**) MH7A cells were transfected with miRNA mimics and stimulated by TNF-α or IL-1β for 1 to 6 h. The cells were harvested, and nuclear protein were extracted. Western blotting of p65 was performed. The expression of histone H3 (H3) is shown as a loading control. (**C**) A densitometric analysis of the data shown in (**B**). Western blotting was performed in quadruplicate. Representative data are shown. Data are expressed as the mean ± SEM. (**D**) MH7A cells were transfected with miRNA mimics and stimulated by TNF-α or IL-1β for 0.5 to 6 h. Cells were subjected to a TransAM assay to evaluate the DNA binding activity of the NF-κB p65 subunit. Assays were performed in triplicate or quadruplicate. Data are expressed as the mean ± SEM. (**E**) MH7A cells were co-transfected with pAP-1-Luc along with miRNA mimics and were then exposed to TNF-α or IL-1β for 6 or 24 h. Cells were then subjected to a luciferase assay to evaluate the activity of AP-1. The luciferase activity was normalized by the number of viable cells and then normalized to the respective values in the vehicle samples. Assays were performed in duplicate, and data are expressed as the mean ± SEM.

**Figure 4 ijms-20-00809-f004:**
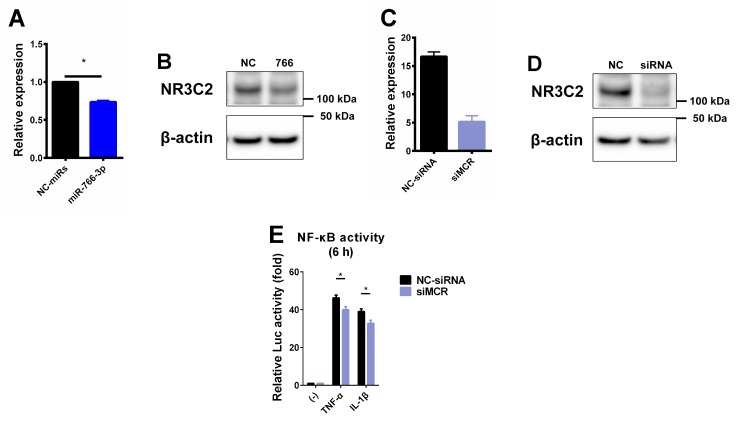
Suppression of NF-κB activation by decreased miR-766-3p-dependent mineralocorticoid receptor expression. (**A**,**B**) MH7A cells were transfected with miRNA mimics. After 48 h, cells were subjected to (**A**) a qPCR or (**B**) western blotting to detect the expression of mineralocorticoid receptor (MCR; NR3C2). (**C**,**D**) MH7A cells were transfected with siRNA scrambles. After 48 h, cells were subjected to (**C**) a qPCR or (**D**) western blotting to detect the expression of MCR. Assays were performed in quadruplicate (**A**,**B**) or duplicate (**C**,**D**), and data are expressed as the mean ± SEM. Representative western blotting data are shown. An asterisk indicates a statistically significant difference (*p* < 0.05). (**E**) MH7A cells were co-transfected with pNF-κB-Luc along with siRNA scrambles and were then exposed to TNF-α or IL-1β for 6 h. Cells were then subjected to a luciferase assay to evaluate the activity of NF-κB. The luciferase activity was normalized by the number of viable cells and then normalized to the respective values in the vehicle samples. Assays were performed in sextuplicate, and data are expressed as the mean ± SEM. An asterisk indicates a statistically significant difference (*p* < 0.05). (**F**) MH7A cells were transfected with miRNA mimics and stimulated by TNF-α or IL-1β for 1 to 6 h. The cells were harvested, and nuclear protein was extracted. Western blotting of p65 was performed. The expression of histone H3 (H3) is shown as a loading control. (**G**) A densitometric analysis of the data shown in (**F**). Western blotting was performed in quadruplicate. Representative data are shown. Data are expressed as the mean ± SEM. (**H**) MH7A cells were transfected with miRNA mimics and stimulated by TNF-α or IL-1β for 0.5 to 6 h. Cells were subjected to a TransAM assay to measure the DNA binding activity of the NF-κB p65 subunit. Assays were performed in quadruplicate. Data are expressed as the mean ± SEM. (**I**) MH7A cells were transiently transfected with pNF-κB-Luc vector. After overnight incubation twice, cells were pre-treated eplerenone (EPL; 5 μM) for 30 min and treated with aldosterone (ALD; 100 nM), TNF-α, and IL-1β for 6 h. Cells were then subjected to a luciferase assay to evaluate the activity of NF-κB. The luciferase activity was normalized by the number of viable cells and then normalized to the respective values in the vehicle samples. Assays were performed in quadruplicate, and data are expressed as the mean ± SEM.

**Figure 5 ijms-20-00809-f005:**
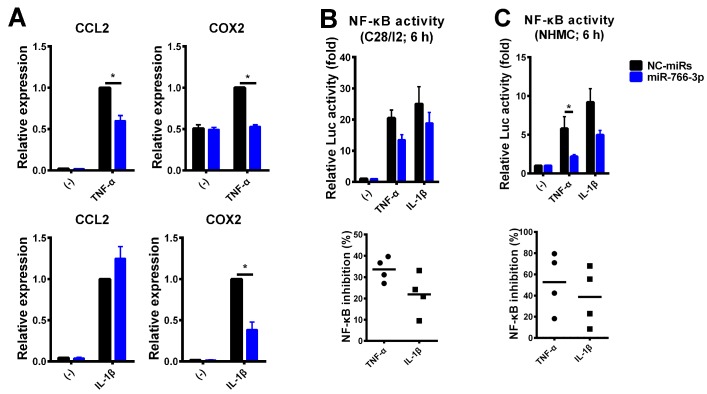
Generalizability of the suppression of inflammatory stimuli in miR-766-3p-transfected cells. (**A**) NHMCs were transfected with miRNA mimics (30 nM) and exposed to TNF-α or IL-1β for 24 h. The expression of the indicated genes was examined by a qPCR. (**B**,**C**) C28/I2 cells (**B**) and NHMCs (**C**) were co-transfected with pNF-κB-Luc along with miRNA mimics (C28/I2: 5 nM, NHMC: 30 nM). After incubation, cells were treated with TNF-α or IL-1β for 6 h and subjected to a luciferase assay to evaluate the activity of NF-κB. The luciferase activity was normalized by the number of viable cells and then normalized to the respective values in the vehicle samples (upper bar graphs). The lower scatterplots show the frequency of NF-κB inhibition by miR-766-3p. Assays were performed in quadruplicate. Data are expressed as the mean ± SEM. Asterisks indicate statistically significant differences (*p* < 0.05).

**Table 1 ijms-20-00809-t001:** Changes in the MicroRNA (miRNA) expression in rheumatoid arthritis (RA) patients after abatacept treatment.

miRNA Name (miRBase ver.17)	ID	Detection Number (Before)	Detection Number (After)	*p* Value	Fold
					Before/After
**Down-regulated miRNAs compered to before treatment**					
hsa-miR-4459	MIMAT0018981	10	10	0.043	−1.99
**hsa-miR-625***	**MIMAT0004808**	**10**	**9**	**0.002**	−1.80
hsa-miR-146a	MIMAT0000449	5	8	0.041	−1.73
**hsa-miR-4505**	**MIMAT0019041**	**10**	**10**	**0.021**	−1.72
hsa-miR-4520a-5p, hsa-miR-4520b-5p	MIMAT0019235, MIMAT0020299	4	4	0.020	−1.66
**hsa-miR-4739**	**MIMAT0019868**	**10**	**10**	**0.004**	−1.58
hsa-miR-3616-3p	MIMAT0017996	7	8	0.038	−1.42
**hsa-miR-766**	**MIMAT0003888**	**10**	**10**	**0.009**	−1.38
hsa-miR-4442	MIMAT0018960	10	10	0.043	−1.28
hsa-miR-149	MIMAT0000450	7	5	0.026	−1.28
hsa-miR-4675	MIMAT0019757	9	10	0.033	−1.27
hsa-miR-3189-3p	MIMAT0015071	5	3	0.031	−1.23
hsa-miR-3197	MIMAT0015082	10	10	0.050	−1.22
hsa-miR-4716-5p	MIMAT0019826	9	7	0.050	−1.19
hsa-miR-4508	MIMAT0019045	10	10	0.039	−1.16
hsa-miR-409-3p	MIMAT0001639	4	4	0.025	−1.03
hsa-miR-1288	MIMAT0005942	3	8	0.031	−1.03
**Up-regulated miRNAs compered to before treatment**					
hsa-miR-3663-5p	MIMAT0018084	8	6	0.038	1.05
hsa-miR-3713	MIMAT0018164	5	5	0.048	1.08
hsa-miR-760	MIMAT0004957	10	10	0.037	1.16
hsa-miR-3202	MIMAT0015089	5	5	0.006	1.17
hsa-miR-4317	MIMAT0016872	4	5	0.038	1.20
hsa-miR-657	MIMAT0003335	7	6	0.043	1.20
hsa-miR-4728-3p	MIMAT0019850	9	9	0.014	1.24
hsa-miR-4651	MIMAT0019715	10	10	0.031	1.33
hsa-miR-4640-5p	MIMAT0019699	10	10	0.027	1.37
hsa-miR-3648	MIMAT0018068	10	10	0.048	1.42
hsa-miR-762	MIMAT0010313	10	10	0.040	1.42
hsa-miR-4669	MIMAT0019749	9	9	0.037	1.43
hsa-miR-1193	MIMAT0015049	9	10	0.046	1.46
hsa-miR-1915	MIMAT0007892	10	10	0.048	1.47
**hsa-miR-1225-5p**	**MIMAT0005572**	**9**	**10**	**0.015**	1.49
hsa-miR-124	MIMAT0000422	8	5	0.026	1.49
**hsa-miR-4299**	**MIMAT0016851**	**10**	**10**	**0.004**	1.58
hsa-miR-3942-3p	MIMAT0019230	4	3	0.037	1.62
hsa-miR-3622a-3p	MIMAT0018004	7	5	0.025	1.66
**hsa-miR-1203**	**MIMAT0005866**	**9**	**10**	**0.032**	1.79
hsa-miR-1231	MIMAT0005586	6	7	0.047	1.80
hsa-miR-4326	MIMAT0016888	6	10	0.047	1.98
hsa-miR-2276	MIMAT0011775	6	6	0.025	2.00
hsa-miR-4664-3p	MIMAT0019738	6	8	0.044	2.26
**hsa-miR-4259**	**MIMAT0016880**	**10**	**9**	**0.012**	2.43

Plasma was collected from RA patients (Table 2) before and three months after treatment with abatacept. Plasma RNA was isolated and then subjected to a miRNA array analysis. miRNAs with significantly altered expression levels, as determined by a paired *t*-test, are listed. The bold miRNAs are included in Figure 1.

**Table 2 ijms-20-00809-t002:** Clinicopathological characteristics of the RA patients used for the miRNA array.

Characteristics	Before	3 Months After *	1 Year After
Number of patients	10	9	10
Sex, male/female	2/8	2/7	2/8
Age (years)	54.7 ± 13.4	55.9 ± 13.7	54.7 ± 13.4
Disease duration (months)	11.6 ± 7.2	12.0 ± 7.5	11.6 ± 7.2
ESR (mm/h)	39.4 ± 24.1	24.3 ± 16.4	20.8 ± 12.4
C-reactive protein (mg/L)	1.19 ± 1.45	0.30 ± 0.61	0.23 ± 0.39
MMP3 (ng/mL)	166 ± 137	212 ± 152	143 ± 97
DAS28-CRP	4.20 ± 1.15	2.66 ± 0.86	1.94 ± 0.67
Remission (<2.3)	0	3	7
low disease activity (<2.7)	0	2	2
moderate disease activity (2.7–4.1)	5	4	1
high disease activity (>4.1)	5	0	0

* One patient’s data was missing. All values are reported as the mean ± standard deviation. ESR, erythrocyte sedimentation ratio; MMP3, matrix metalloproteinase-3; DAS28-CRP, Disease Activity Score 28 joint count with C-reactive protein.

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
