# Peer review of "MicroRNA-766-3p Contributes to Anti-Inflammatory Responses through the Indirect Inhibition of NF-κB Signaling"

_ijms, 2019, doi:10.3390/ijms20040809_

Round 1

Reviewer 1 Report

In this study, the authors investigated the function of microRNA-766-3p in autoimmune inflammatory disease and related cell lines. The data suggested that microRNA-766-3p plays an anti-inflammatory role by reducing the NF-κB activity, probably through down regulating MCR. In general, this manuscript is very interesting and the results are relatively convincing. Some questions will be asked by readers concerning its acceptance. I listed the suggestions in the following text.

1)    In Figure 1, eight miRNAs, which all from abatacept treatment in RA patients, have been tested, why only microRNA-766-3p has anti-inflammatory function? Do you have any ideas?

2)    What is the possible target (s) of microRNA-766-3p? This point should be discussed carefully.

3)    Please remake Table 2 to keep the table intact and the format consistent

4)    It is better to show the molecular marker in all immunoblots.

5)    The blot pattern of H3 in Figure 3B and 4F seems to be different. There are increased H3 signals in all the “NC” lanes. Why?

Author Response

Response 1:

In this study, we hypothesized that some of the blood-circulating miRNAs that fluctuate before and after abatacept treatment in RA patients, have a physiological function and began verification efforts.

First, we analyzed the plasma miRNA levels using a miRNA array analysis. In the primary screening, we selected miRNAs with a significantly altered expression because they met the following criteria: 1) large variation in expression, 2) stable array results (stable number of detected specimens), and 3) able to be measured. In this way, we obtained the eight miRNAs shown in Figure 1.

At the secondary screening, we restricted our miRNAs for investigation to those that control the inflammatory response, resulting in the identification of hsa-miR-766-3p. Of note, the inhibition of the inflammatory response was observed not only with miR-766-3p but also with other miRNAs (Figure 1).

For the convenience of the experiment, we first selected these eight miRNAs in this study. Therefore, other anti-inflammatory miRNAs may be present in Table 1.

In addition, that we were unable to find any miRNA that promoted the inflammatory response was an unexpected finding. However, we were able to verify that some blood-circulating miRNAs did indeed exert functions.

Response 2:

In addition to MCR, BAX (Liang et al. Cell. Physiol. BIochem. 2013), SIRT6 (Sharma et al. J. Biol. Chem. 2013) (page 9, line 257 to 260), CYP11B2 (Maharjan et al. Physiol. Genomics 2014, 46, (24), 860-865.), SOX6 (Li et al. Onco Targets Ther. 2015, 8, 2981-2988.) and MDM4 (Wang et al. Oncotarget 2017, 8, (18), 29914-29924.) have been reported to be miR-766-3p targets. We searched for miRNA targets using databases such as TargetScan and identified RIPK1, which may be directly involved in NF-κB signaling. However, we were unable to confirm the knockdown of RIPK1 by miR-766-3p in our experiment.

Response 3:

As suggested, Table 2 was revised.

Response 4:

As another reviewer suggested, we revised all immunoblots.

In addition, the molecular markers are now shown in all immunoblots.

Response 5:

We reviewed all of the immunoblot experiments, it was not a specific change by knockdown. Indeed, miR-766-3p and siMCR do not reduce the expression of H3 (Appendix; Word File).

Reviewer 2 Report

In this manuscript Hayakawa and Coworkers investigated on hsa-miR-766-3p anti-inflammatory effects. The Authors observed that hsa-miR-766-3p indirectly reduces the mineralocorticoid receptor (MCR; NR3C2) expression in MH7A cells and contributes to suppress the inflammatory responses. Herein, the Authors demonstrate that miR-766-3p reduces NR3C2 expression and probably this event is crucially involved NF-kB activation. However, it is known that NR3C2 is a potential miR-766-3p target and even if Hayakawa at al demonstrates that hsa-miR-766-3 contributes to reduce TNF-induced inflammatory response and is involved in the suppression of cytokine-induce NF-kB activation the present work required concerns before publication.

-       In the figure 3A, the Authors observed a significative inhibition of NF-kB activation in MH7A cells stimulated with TNF-a or IL-1 b for 6 or 24 hours in presence of miR-766-3p. What about NF-kB activity levels 1 or 3 hours after the treatment with TNF-a or IL-1b in presence of miR-766-3p or NC-miRs?

-       In the figure 3 legend is reported a formazan assay (page 6, line 157), but in the figure no pictures represent the assay mentioned in the legend.

-       Western blot in figure 3B is not well normalized. Please change the blot.

-       Western blots in figure 4 are too cropped and blots reported in figure 4B and 4D should be reported in a single blot as well as the densitometric analysis. Furthermore, statistical analysis is missing in figure 4C since siRNA scramble seems silence NR3C2 expression.

-       Again, in the figure 4 legend is reported a formazan assay (page 8, line 202 and 216), but in the figure no pictures represent the assay mentioned in the text.

-       Densitometric analysis reported in figure 4G is not representative of western blot in figure 4F. Please revised the western blot or densitometric analysis.

-       In material and methods is reported formazan assay, but none pictures are reported in the manuscript.

-       Manuscript title describes the anti-inflammatory MicroRNA-766-3p contribute by the inhibition of NF-kB signaling, but the Authors states: “We showed that microRNA-766-3p indirectly reduce the activation of NF-kB” (page 1, lines 27-28). Please change title.

-       The Authors suggest that the suppression of inflammatory responses is exerted when miR-766-3p is extracellulary present in large amounts and is taken into the cell. Furthermore, they hypothesize that mesenchymal stromal cells could be a potential source of miR-766-3p. Did the Authors think that extracellular final concentration at least of 5 nM is reached in the plasma or in tissues under physiopathological conditions?

Author Response

Response 1:

We previously examined the suppression of the inflammatory response at an early phase in order to identify the target of miR-766-3p. Our results showed a tendency toward suppression with stimulation with TNF-α and IL-1β for 1 h in presence of miR-766-3p. Therefore, we suspect that miR-766-3p is not targeted for de novo gene expression by inflammatory stimuli. Indeed, the present findings suggest that knockdown of preexisting NR3C2 by miR-766-3p is important.

Response 2, 5:

The results of the luciferase activity in Figure 3A, 3E, 4E, 4I were normalized with in viable cells. The formazan assay was performed to assess the presence of viable cells. To avoid any further confusion, we deleted mention of this from the text.

Response 3, 6:

As suggested, we revised the western blot in Figure 3B and 4F. However, despite these changes, H3 remained poorly normalized. Meanwhile, since miR-766-3p and siMCR do not reduce the expression of H3 (Appendix 1; Word File), so this may be a technical problem.

Response 4:

Figures 4B and 4D were revised to show the same immunoblot (Appendix 2; Word File). However, we decided to refrain from combining them in order to explain in manuscript the knockdown of MCR by miRNA and siRNA.

In addition, because the assay for Figure 4C was performed in duplicate, a statistical analysis was not performed. The siRNA scrambles were commercially available and were known to be sufficiently effective.

Response 7:

We assessed the presence of viable cells with a formazan assay using a Cell Counting kit-8 (Dojindo; https://www.dojindo.eu.com/store/p/456-Cell-Counting-Kit-8.aspx). The data obtained from the formazan assay are shown in Figure 2D. In addition, as mentioned above (Response 2, 5), these findings were used for data normalization (Figure 3A, 3E, 4E, 4I).

Response 8:

As suggested, we revised the title of this study to “MicroRNA-766-3p contributes to anti-inflammatory responses through the indirect inhibition of NF-κB signaling”.

Response 9:

First off, we apologize for our error, as the expression should be mesenchymal “stromal” cells instead of mesenchymal “stem” cells. Thank you for noting our mistake. Furthermore, there were some discrepancies in the conclusions of the Yan et al. study (Sci Rep. 2016; reference [30]) and our own. We therefore replaced this reference with the suitable (Chen et al. J Immunol. 2018) based on our discussion.

In our study, miR-766-3p partially inhibited the inflammatory responses even below 5 nM (Figure 2A). However, while the transfection efficiency of MH7A cells was very good, how many picomoles of miR-766-3p were transferred into each cell is unclear. The experiments by Chen et al. (J Immunol. 2018) showed that exosomes produced by mesenchymal stem cells lead to the suppression of collagen-induced arthritis but did not mention the absolute concentration of miRNAs. Furthermore, to our knowledge, no equation for calculating the absolute concentration of miRNAs in tissues or plasma has been proposed. However, the amount of miRNA taken up by the cell is believed to be important. In blood, circulating miRNAs are packaged in microparticles or associated with RNA-binding proteins. These forms can exert physiological effects even at a low miRNA concentration. However, in the present study, we conducted general transfection experiments, and did not mimic exact in vivo conditions. Therefore, we do not have an accurate answer regarding whether or not 0.1 to 5 nM is pathophysiological concentration and cannot deny the possibility of the concentration being quite high under pathophysiological conditions.

Round 2

Reviewer 2 Report

No more concerns.